

# Effects of temperature and size class on the gut digesta microbiota of the sea urchin *Tripneustes ventricosus*

Ruber Rodríguez-Barreras[1], Eduardo L. Tosado-Rodríguez[2], Anelisse Dominicci-Maura[2] and Filipa Godoy-Vitorino[2]

[1] Department of Biology, University of Puerto Rico at Bayamón, Bayamón, PR, USA
[2] Microbiology and Medical Zoology, University of Puerto Rico, School of Medicine Medical Sciences Campus, San Juan, PR, USA

## ABSTRACT

**Background:** Understanding the dynamics of the gut microbiota in sea urchins is crucial for comprehending the ecological balance in marine ecosystems. The gut microbiota plays a vital role in nutrient metabolism, immune system modulation, and pathogen protection. The microbial composition and dynamics of naturally occurring sea urchin *Tripneustes ventricosus* have yet to be thoroughly investigated. We hypothesized the gut microbiota of *T. ventricosus* in the Caribbean, varies across life stages and seasons.

**Methods:** Thirty-six naturally occurring large individuals and six small individuals (42 animals) were collected from shallow waters on the northeastern coast of Puerto Rico in February and August of 2019. The fecal pellet's microbiota was characterized by sequencing V4 region of the 16S rRNA gene.

**Results:** We found significant differences in the composition of fecal pellet microbiota between seasons and life stages. Phylum Bacteroidota had greater relative abundance in August, while Firmicutes was more dominant in February. *Propionigenium* and *Roseimarinus* had greater relative abundance in August, while *Candidatus* Hepatoplasma, and *Kistimonas* had greater relative abundance in February. Differences in the gut digest microbiota were not found between small and large urchins, but small urchins displayed a slightly higher diversity and dominance of Bacteroidota and Proteobacteria, while large urchins exhibited a greater relative abundance of Fusobacteria and Desulfobacterota. However, the genera *Ferrominas* and *Propionigenium* counts were significantly lower in small individuals.

**Discussion:** This is the first report for this species in the Caribbean region and adds to our comprehension of the microbiota of the white sea urchin across collection periods and size classes, highlighting the dynamic nature of the gut microbiota.

# INTRODUCTION

Only approximately 1% of the global prokaryotic biodiversity has been successfully cultured in laboratory conditions using conventional methods (*Schleifer, 2004*; *López-García & Moreira, 2008*). The challenge of understanding the culturability of many

Corresponding authors
Ruber Rodríguez-Barreras,
ruber.rodriguez@upr.edu
Filipa Godoy-Vitorino,
fgodoyvitorino@gmail.com

bacterial taxa has been addressed by employing alternative technological approaches. The advancement of molecular techniques has significantly enhanced our comprehension of the prokaryotic biodiversity (*Vartoukian, Palmer & Wade, 2010*). Currently, one of the most widely used methods for characterizing prokaryotic communities in marine invertebrates like echinoderms, involves culture-independent identification through 16S ribosomal RNA gene sequencing (*Hakim et al., 2015*; *Pagán-Jiménez et al., 2019*; *Hastuti, Fatma & Tridesianti, 2023*). The number of studies on naturally occurring wild sea urchins has also increased in the last decades (*Hakim et al., 2019*; *Faddetta et al., 2020*; *Ketchum et al., 2021*; *Rodríguez-Barreras, Tosado-Rodríguez & Godoy-Vitorino, 2021*).

Over the last decade, sequencing the 16S rRNA gene through amplicon sequencing has emerged as a quick and affordable method for analyzing microbiota composition and diversity including the host gut microbiota associated with the digestive system, across animal phyla ranging from invertebrates to vertebrates (*Lee & Hase, 2014*; *Grinevich et al., 2024*). Recently, there has been a significant increase in knowledge about how environmental elements can impact the makeup and behavior of prokaryotic communities (*Ward et al., 2017*; *Fontaine, Novarro & Kohl, 2018*; *Sepulveda & Moeller, 2020*; *Traving et al., 2021*).

Sea urchins have been widely used in host-microbiota studies among marine invertebrates (*Hakim et al., 2019*; *Schwob et al., 2020*; *Miller et al., 2021*). The biological fitness of echinoderms, including sea urchins, heavily depends on the symbiotic relationship with their microbiota, which performs essential functions to the organism's resilience (*Ho et al., 2016*; *Carrier & Reitzel, 2018*; *Schuh et al., 2020*). Echinoderms rely on diverse microorganisms within their bodies to carry out vital processes such as nutrient metabolism, immune system modulation, and protection against pathogens (*Schuh et al., 2020*). The intricate interdependence between echinoderms and their microbiota underscores the critical role of symbiotic interactions in their host survival and performance (*Carrier & Reitzel, 2019*, *2020*; *Carrier et al., 2021*). For example, a recent study stated that symbiosis in the sea urchin *Brisaster townsendi* plays different roles in the host nutrition (*Ziegler et al., 2020*), while another study found the occurrence and importance of a photosynthetic bacteria as a nutrition supporter in the seastar *Mithrodia clavigera* (*Galac, Bosch & Janies, 2016*).

Changes in microbial communities are often associated with changes in environmental conditions (*Dang et al., 2023*; *Zeng et al., 2023*). Temperature has emerged in the literature as a prominent abiotic factor and a reliable predictor, driving significant shifts in prokaryotic taxa within thermally variable habitats (*Ketchum et al., 2021*). The gut microbiota plays a critical role in host phenotypic plasticity (*Kolodny & Schulenburg, 2020*) in many ways through morphological changes, physiological adaptations, behavioral responses, and life history strategies (*Gotthard & Nylin, 1995*). Modifications in environmental temperature can result in significant changes to the gut microbiota diversity in echinoderms (*Gao et al., 2014*; *Brothers et al., 2018*). Conversely, the microbiota,

through their metabolites, can serve as a feedback mechanism, enhancing host plasticity in thermoregulatory mechanisms (*Khakisahneh et al., 2020*).

Another driving factor explored is the relationship between aging and host microbiota (*Kawamoto & Hara, 2024*). Traditionally, studies have primarily focused on understanding the consequences of changes in microbiota on nutrition and immune-related processes in both invertebrates and vertebrates (*Clark & Walker, 2018*; *Maynard & Weinkove, 2018*; *Derrien, Alvarez & de Vos, 2019*; *Miró et al., 2020*). However, understanding how prokaryotic taxonomic composition changes with age in marine invertebrates, particularly echinoderms, remains limited. In addition, the dynamic nature of the gut microbiota in response to seasonal changes underscores the adaptability of marine invertebrates to diverse environmental conditions. The dynamic nature of the gut microbiota in response to seasonal changes underscores the adaptability of marine invertebrates to diverse environmental conditions seasonal dynamics of the gut microbiota in marine invertebrates have important implications for marine ecosystems' overall health and resilience (*Ketchum et al., 2021*). Seasonal shifts in microbial communities can influence nutrient cycling, disease resistance, and the overall fitness of the host organisms (*Lee, Wong & Qian, 2009*).

Sea urchins are considered suitable models for microbiota studies due to their anatomical simplicity, ecological importance, ease of collection and maintenance. The White Sea urchin, *Tripneustes ventricosus* (Lamark, 1816), plays a pivotal role in shaping coastal ecosystems, influencing benthic communities through algae grazing, and contributing to nutrient cycling (*Lawrence & Agatsuma, 2007*). The species is considered one of the largest regular echinoids in the western Atlantic and the Caribbean (*Hendler et al., 1995*; *Rodríguez-Barreras, Sabat & Calzada-Marrero, 2013*). *T. ventricosus* is characterized by a rapid growth, sexual maturity, and short longevity (*McPherson, 1965*). It usually inhabits back-reef areas dominated by marine flowering plants like *Thalassia testudinum* and *Syringodium filiforme* (*Tertschnig, 1989*; *Hendler et al., 1995*). This sea urchin *T. ventricosus* is primarily herbivorous and plays an important role in the dynamic of seagrass meadows. A dietary characterization of the sea urchin, using gut content analysis by stable isotopes and DNA-metabarcoding, revealed the eukaryotic composition of the ingested material (*Maciá & Robinson, 2008*; *Rodríguez-Barreras et al., 2016*, *2020*).

While the gut and epibiotic microbiota in large urchins have been studied (*Rodríguez-Barreras, Tosado-Rodríguez & Godoy-Vitorino, 2021*; *Rodríguez-Barreras et al., 2023*), there is still a gap in our understanding about the dynamic of the gut prokaryotic community between size classes. Additionally, considering the rise of ocean temperatures across different seasons (*Williams, Williams & Logan, 2023*), it becomes critical to understand how the host microbiota responds to seasonal changes in temperature. Therefore, the objectives of this study were (1) comparisons of the gut microbiota in *T. ventricosus* during February (low temperature) and August (high temperature), and (2) comparing the gut microbiota between small and large size classes. Our hypothesis states that the gut microbiota will likely change between the two collection periods and between individual size class.

## MATERIALS AND METHODS

### Study site and sample collection

This study was conducted at three shallow-water seagrass meadows of Puerto Rico's northeastern coast. Sites from East to West were Cerro Gordo in Vega Baja (CGD-18°29′ 06.0″N; 66°20′20.1″W), Isla de Cabra in Toa Baja (ICB-18°28′26.6″N; 66°08′18.5″W), and Punta Bandera in Luquillo (PTB-18°23′16.0″N; 65°43′05.2″W). The Department of Natural and Environmental Resources of Puerto Rico approved a collection permit for this study (permit number: DRNA-2019-IC-003). Limitation in the number of collected individuals is due to the permit limitation. All sites have a well-developed seagrass meadow dominated mostly by the flowering plants *Thalassia testudinum* and *Syringodium filiforme*, with an average between 0.5 and 1.5 m depth. Additional site description and map are available in *Rodríguez-Barreras, Tosado-Rodríguez & Godoy-Vitorino (2021)*. The abiotic parameters (salinity, water temperature, and pH) were measured using a Pro-2030 quality meter (Xylem Inc., Washington, DC) during February and August of 2019. Each abiotic parameter was calculated based on the average of five repetitive measures. Abiotic factors varied between February and August of 2019 (Table S1). A Kruskal-Wallis rank sum tests were conducted to assess potential differences in temperature, salinity, and pH between February and August. Normality and homogeneity of variance were previously tested using the 'car' package (*Harrell, 2021*). A non-parametric Mann-Whitney test was run to compare differences in horizontal test diameter between small and large size classes. All tests were run in R version 4.3.2 with a significance level (*p*-value) of 0.05.

We randomly selected six large of the sea urchin *Tripneustes ventricosus* during February and August by site. We also collected six small individuals only in Isla de Cabra for a total of 42 echinoids. We classified a large urchin any individual with a horizontal test diameter greater than 70.1 mm. This threshold was based on the average size of both groups (Table S2), not in physiological maturity (*McPherson, 1965*). Measures were taken with a caliper (error ± 0.05 mm). *T. ventricosus* gut microbiota data for February of 2019 was taken from *Rodríguez-Barreras, Tosado-Rodríguez & Godoy-Vitorino (2021)* and those of August 2019 (small and large individuals) are being reported here. Site collections were conducted on various days within the same month for each site to prevent the potential mixing of individuals from different sites. Collected specimens were temporarily placed in a foam cooler filled with seawater, equipped with an air battery-supplied pump, for transportation to the laboratory facility.

### Sample processing

A chemical method was used for induced euthanasia as described in the approved IACUC protocol [A-5301118]. Once in the laboratory, each individual was placed inside a 100 mL glass beaker with seawater for at least 10 min until it was attached to the surface, and then sedated by adding 25 mL of a 20 mM Magnesium Chloride (MgCl₂) hexahydrate solution. This chemical procedure is commonly used in marine invertebrates (*Arafa, Sadok & Abed, 2007*; *Doerr & Stoskopf, 2019*; *Wahltinez et al., 2021*). Sea urchins were completely detached from the wall of the beaker after the anesthesia effect. After that, individuals were

relocated into a metal tray and exposed to ultra-low temperature of −80 °C for 10 min before dissecting. Lifeless individuals were placed in a metal tray and carefully opened with an equatorial cut around the oral membrane using a flame-sterilized scissor, avoiding damage to the digestive tract (*Whalen, 2008*). The gut was cut, opened, and fecal pellets transferred with sterilized tweezers to a Petri dish. Next, fecal pellets were put to 2 mL microtubes and placed in a freezer at −80 °C before DNA extraction. This procedure focuses on the isolation of the bacterial community associated with gut digesta, specifically excluding tissue-associated. These procedures were approved by the University of Puerto Rico Medical Sciences IACUC protocol (A-5301118).

## DNA extraction, amplification, and sequencing

To isolate genomic DNA from gut fecal pellets, we employed the QIAGEN PowerSoilTM kit (QIAGEN LLC, Germantown Road, Maryland, USA) with some modifications to the manufacturer's instructions. Gut fecal pellets were homogenized using a PowerLyzer homogenizer for 2 min at room temperature, running at 3,000 r.p.m. The elution step included incubating the eluent in 100 μl of sterile PCR water, pre-heated to 65 °C for 5 min, followed by a final centrifugation step. The concentration of the purified DNA extracts was determined using the Qubit® dsDNA HS Assay Kit with the Qubit® Fluorometer at room temperature, ranging from 5–100 ng/μl (Waltham, Massachusetts, U.S.).

During the 16S library preparation, the DNA extracted from gut fecal pellets was standardized to 4nM. To amplify the V4 hypervariable region of the 16S ribosomal RNA marker gene, we utilized universal bacterial primers: 515F (5′GTGCCAGCMGCCGC GGTAA3′) and 806R (5′GGACTACHVGGGTWTCTAAT3′). The amplification was conducted following the protocols provided by the Earth Microbiome Project (http://www.earthmicrobiome.org/emp-standard-protocols/16s/) (*Caporaso et al., 2012*), using previously established conditions (*Abarca et al., 2018*). The 16S rRNA amplicons were sequenced using Illumina MiSeq Reagent kit with a 2 × 250 bp setup (V4 region). The resulting 16S-rRNA sequences were submitted to QIITA (*Gonzalez et al., 2018*) under the Bioproject ID 12668; the raw sequences are publicly accessible in the European Nucleotide Archive under ENA Projects PRJEB40117 and ERP123720.

The dataset, publicly deposited in 2021, includes sequence data from various species and approaches related to different projects. We used the published February data of large *Tripneustes ventricosus* from *Rodríguez-Barreras, Tosado-Rodríguez & Godoy-Vitorino (2021)* to compare with our new unreported data for small and large individuals of the same species collected in August of 2019 (reported here).

## QC processing

The initial 16S rRNA raw FASTQ sequence files and their associated metadata information were deposited in QIITA, as described in *Gonzalez et al. (2018)*. The demultiplexed files were raw read pre-processing using split libraries FASTQ with default parameters and a Phred offset of 33, as implemented in QIIMEq2 1.9.1 (*Bolyen et al., 2019*). The sequences were initially trimmed to a length of 250 bp, and then the deblurring workflow (deblur

1.1.0) was applied (*Gonzalez et al., 2018*; *Bolyen et al., 2019*). The resulting species table was downloaded for further analyses using a locally run version of QIIME2 (*Bolyen et al., 2019*). To assign taxonomy, we used the Silva 138 reference database, specifically targeting the 515F/806R region of the sequences, with a minimum similarity threshold set at 99% (*Quast et al., 2012*). The Naive Bayes trained classifier for this database was obtained from https://docs.qiime2.org/2023.2/data-resources/ and employed for taxonomy classification using the sklearn tool in QIIME2 (*Bokulich et al., 2018*). Amplicon sequence variants (ASVs) with fewer than five reads and sequences, those matching chloroplasts, mitochondria, and taxonomically unassigned sequences, were excluded from subsequent analyses. For the comparison between seasons, we performed rarefaction at a level of 17,000 reads per sample, while for the comparison between ages, all samples were rarefied to 4,500 reads per sample. The sample distribution across sites and seasons consisted of 11 samples from ICB, 12 samples from CGD, and another 12 samples from PTB, resulting in a total of 35 samples of large urchins; small urchins were not included in these analyses (Table S2). This analysis was adjusted for the sample site to mitigate bias introduced by co-variables.

## Analyses of microbial communities and statistical testing

The reads were used for an alignment using MAFFT, in which phylogenetically uninformative or ambiguously aligned columns will be removed (masked). The resulting masked alignment will infer a phylogenetic tree with "qiime phylogeny align-to-tree-mafft-fasttree" in QIIME2. This step is important to calculate alpha diversity index "faith_pd" (*Faith, 1992*) with "qiime diversity alpha-phylogenetic" plotted as rarefaction curves. Additionally, we calculated observed features as well as the Shannon index (*Shannon & Weaver, 1949*). Statistical analyses for alpha diversity were done using "qiime diversity alpha-group-significance" script in QIIME2, which uses a non-parametric t-test with Monte Carlo permutations (*Bolyen et al., 2019*). Taxonomic bar plots for phylum and genus were also generated using Microbiome Analyst 2.0 (*Lu et al., 2023*).

Beta diversity within our categories was calculated using DEICODE plugin in QIIME2 (*Martino et al., 2019*). DEICODE allows us to identify significant inter-community niche features and visualize them in compositional biplots (*Martino et al., 2019*). The resulting ordination file was modeled using the "qiime emperor biplot" script in QIIME2 (*Bolyen et al., 2019*). Robust Aitchison principal component analysis (PCA biplots) serve as visualizations, illustrating arrows that correspond to the specific feature (taxonomically characterized) and responsible for group clustering (*Martino et al., 2019*). Arrows respond to Euclidian distance from the origin, and their size indicates the strength of the relationship of that ASV to the community composition and grouping. The QIIME2 Emperor biplot script selects the top feature arrows based on the magnitude of all the dimensions, while the largest value in each matrix does the scaling of the arrows.

To compare the ranked beta diversity distances across the different variables, we used a Bray-Curtis dissimilarity table. We conducted PERMANOVA analyses using the adonis function with stratification from the vegan package in R to compare seasonal variations within the same site (*Oksanen et al., 2014*). These tests were conducted using the "qiime

diversity beta-group-significance" script in QIIME2, with 999 permutations (*Bolyen et al., 2019*; *Martino et al., 2019*).

Using a linear model, we applied MaAsLin2 in Microbiome Analyst 2.0 to conduct a multivariable association analysis between taxa and features of interest (season and size classification) (*Lu et al., 2023*; *Mallick et al., 2021*). To minimize statistical bias, this analysis was corrected for sample sites. The key metadata variables considered in the study were 1- Size-class (small and large), 2- water temperature (February and August), and 3- sample site (CGD, ICB, and PTB). We exported the Maaslin2 results to create a volcano plot using the VolcaNoseR tool (*Goedhart & Luijsterburg, 2020*). In the plot, significant bacterial taxa were highlighted based on a threshold of $p < 0.05$ (1.3 on -log10 scale) and a minimum fold change of 1.5, as per Goedhart & Luijsterburg recommendations. For enhanced clarity in the plot, we transformed the $p$ to a -log10 scale to directly correlate the scale with increasing significance. We used linear discriminant analysis (LDA) with LefSe, an algorithm for biomarker discovery that identifies taxa characterizing the differences between the metadata classes (*Segata et al., 2011*).

# RESULTS

## Quality assessment and spatial changes in the gut microbiota among large sea urchins

Abiotic factors exhibited no spatial differences across seasons and collection sites, except water temperature, which changed between seasons ($p = 2.563e{-}06$) (Table S2). In terms of the gut microbiota, after quality control, a total of 2,642,046 high-quality sequence reads remained; 2,316,458 were used in the analyses of all large urchins, while 325,588 corresponded to the six small sea urchins (Table 1, Table S2). The relative abundance at the genus level varied slightly within each site with *Propionigenium* having more relative abundance in PTB (Fig. S1A). The adult gut microbiota remained relatively similar in adult individuals with no significant differences in alpha diversity across sites (non-parametric t-test with Monte Carlo permutations; $p = 0.696$) (Fig. S1B). Alpha diversity was also similar among sites ($p = 0.221$), although ICB and CGD showed slightly higher Shannon index than PTB (Fig. S1C).

## Temperature changes in the gut microbiota among large animals are more significant than among collection sites

Most differences at the phylum level include a higher relative abundance of Bacteroidota in August (average 39.8% in August *vs.* 28.8% in February), while Firmicutes were higher in February (31.9% *vs.* 6% in August (Fig. 1A, Table S3). At the genus level, a higher relative abundance of *Desulfatolea* (~7.4%), *Propionigenium* (23.7%) and *Roseimarinus* (15.9%) e in high-temperature period samples, while *Candidatus* Hepatoplasma (~16.5%), *Fusibacter* (11.9%), and *Roseimarinus* (15.9%) had greater relative abundance in February (Fig. 1B, Table S3). Across seasons, a permutational statistical test based in ASVs confirmed significant dissimilarities in bacterial community composition between seasons (Permanova, $p = 0.001$) (Table S4). Inter-community features were highlighted using DEICODE compositional biplots. The analysis revealed that the genera *Fusibacter* were

**Table 1 Average spatial and seasonal number of reads and OTUs for the 41 samples considered in the analyses of the white sea urchin *Tripneustes ventricosus*.**

| Sites | Time | Number of samples | Average of reads ± SD | Average of ASVs ± SD |
|---|---|---|---|---|
| Cerro Gordo | February | 6 large | 12,119 ± 7,859 | 823.33 ± 482.53 |
| | August | 6 large | 71,743 ± 22,410 | 574.50 ± 234.36 |
| Isla de Cabra | February | 6 large | 58,475 ± 31,904 | 1,295.33 ± 621.09 |
| | August | 6 small | 55,127 ± 19,150 | 652.33 ± 249.51 |
| | August | 5 large | 191,757 ± 203,450 | 1,150.60 ± 736.01 |
| Punta Bandera | February | 6 large | 23,550 ± 15,313 | 637.00 ± 212.12 |
| | August | 6 large | 76,465 ± 35,969 | 557.50 ± 364.09 |

Note:
Reads and ASVs expressed as the average ± standard deviation (SD) after quality control analysis.

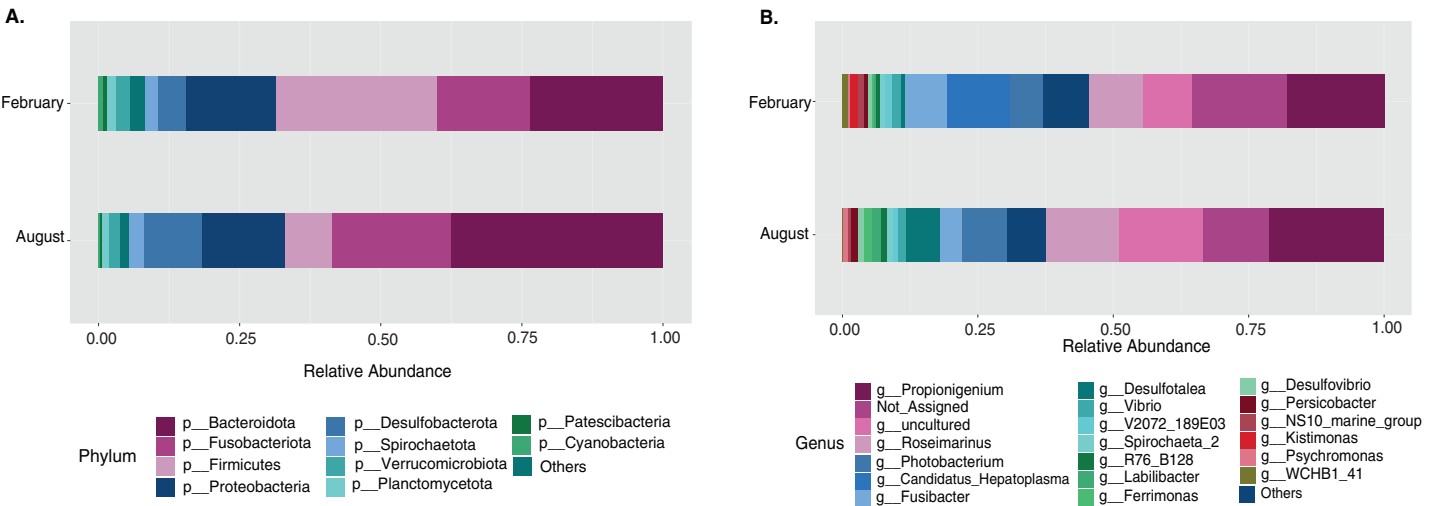

**Figure 1 Temporal bacterial taxonomic distribution in the sea urchin *Tripneustes ventricosus* (February *n* = 18; August *n* = 17).** Taxonomic plots show the average relative abundance at Phyla (A) and genus (B) levels, depicting relative abundances.

dominant in February samples; while in August, in addition to *Propionigenium*, the order Bacteroidales, and the genera *Roseimarinus*, *Fusibacter*, *Desulfotalea*, and *Photobacterium* also dominated. *Propionigenium* were dominant in the two time periods (Fig. 2A). Additionally, alpha diversity analyses revealed significant differences between seasons. There was an increase Shannon diversity in February ($p$ = 0.0022) (Fig. 2B), and in observed features ($p$ = 0.0022), while faith_pd remained similar between seasons ($p$ = 0.390) (Fig. 2D). The seasonal dynamics of the gut microbiota, analyzed using MaAsLin2, identified a total discriminating 176 ASVs at the genus level (Fig. 3), out of which only 12 showed significant differentiation (FC = 1.5; FDR $p$ = 0.05). Specifically, seven taxa exhibited a significant decrease of at least 1.5-fold in February compared to August, including *Desulfotalea*, Sediminispirochaeta, SCGC_AAA286_E23, Marinimicrobia SAR406 clade, SG8_4, *Ferrimonas*, and Woesearchaeales. On the other hand, five other taxa displayed a significant increase in relative abundances of at least 1.5-

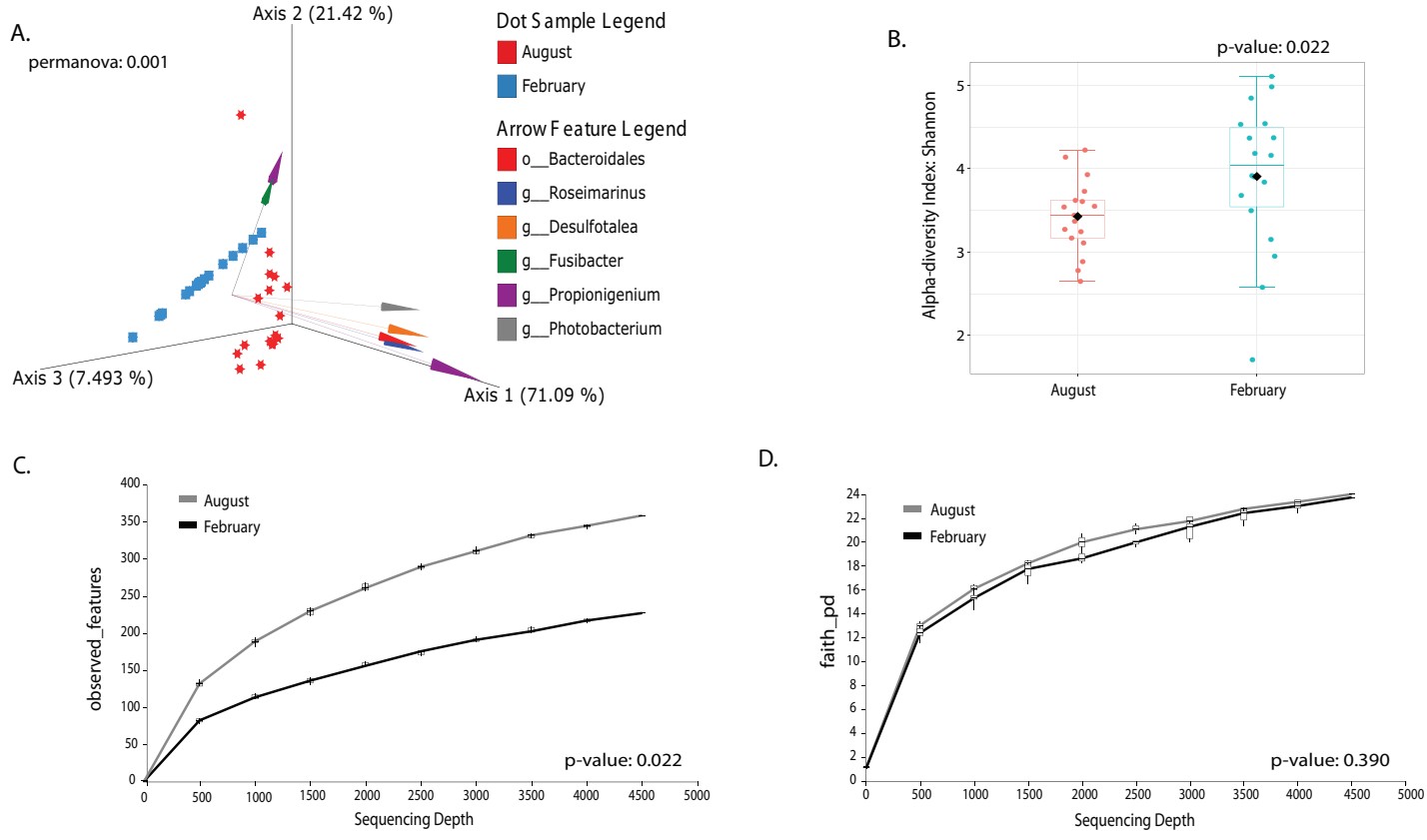

**Figure 2** Temporal beta and alpha diversity analyses of the sea urchin *Tripneustes ventricosus* gut microbiota (February *n* = 18; August *n* = 17). Beta diversity is represented in a 3D PCA biplot as a principal component analysis. Arrows corresponding to the specific feature (taxonomically characterized) and responsible for group clustering are colored. The arrows respond to Euclidian distance from the origin, and their size indicates the strength of the relationship of that ASV to the community composition and grouping (A). Alpha diversity estimates are visualized by Shannon diversity boxplots (B) and rarefaction curves for observed features (C) and Faith's phylogenetic index (D).

fold in February compared to August, namely MSBL3, *Haloferula*, *Cutibacterium*, RF39, and *Candidatus* Hepatoplasma (Fig. 3).

## Changes in gut microbiota linked with size class

The sample size distribution consisted of six small individuals from ICB and 17 large individuals from ICB, CGD, and PBD, collected exclusively during August. Size classes were significantly different in horizontal test diameter (*p* = 0.004) (Table S2). Statistical tests (PERMANOVA strata) confirmed that the bacterial composition did not differ based on size in *T. ventricosus* even when correcting for sample collection site (*p* = 0.789) (Table S3). Small urchin samples displayed a lower relative abundance of Fusobacteria and higher relative abundance of Bacteroidota and Proteobacteria than large individuals at phylum level (Fig. 4A). At the genus level, *Propionigenium* had greater relative abundance in large urchins, while *Photobacterium* and *Roseimarinus* had greater relative abundance in small urchins (Fig. 4B).

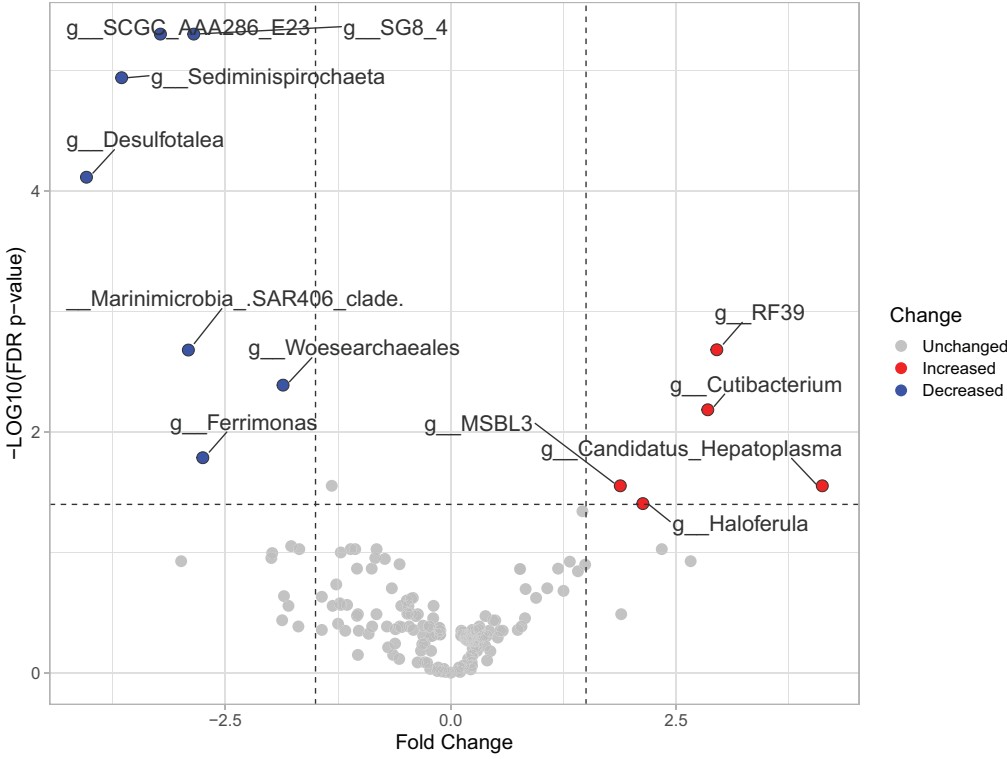

**Figure 3 Volcano plot based on MaAsLin2 analysis comparing taxa between February and August in the sea urchin *Tripneustes ventricosus*.** In the plot, red dots represent bacterial genera significantly had greater relative abundance (FC ≥ |1.5| and $p$ ≤ 0.05) in winter than in summer. Conversely, blue dots indicate significantly reduced genera (FC ≥ |1.5| and $p$ ≤ 0.05) in winter compared to summer. Grey dots represent non-significant features. The plot's X-axis represents the fold change between the two seasonal groups on a log2 scale. At the same time, the Y-axis displays the negative log10 of the $p$-values resulting from the statistical test conducted for the comparison.

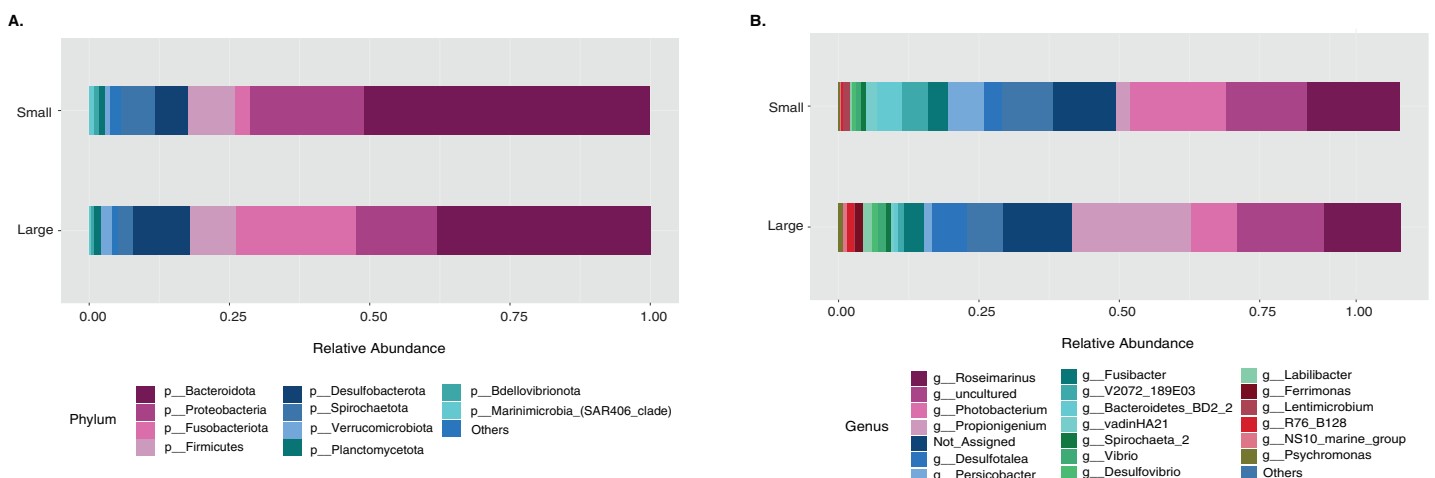

**Figure 4 Bacterial taxonomic distribution according to size classification in the sea urchin *Tripneustes ventricosus*.** Taxonomic plots show the average relative abundance at Phyla (A) and genus (B) levels computed for $n$ = 6 small and $n$ = 35 large animals.

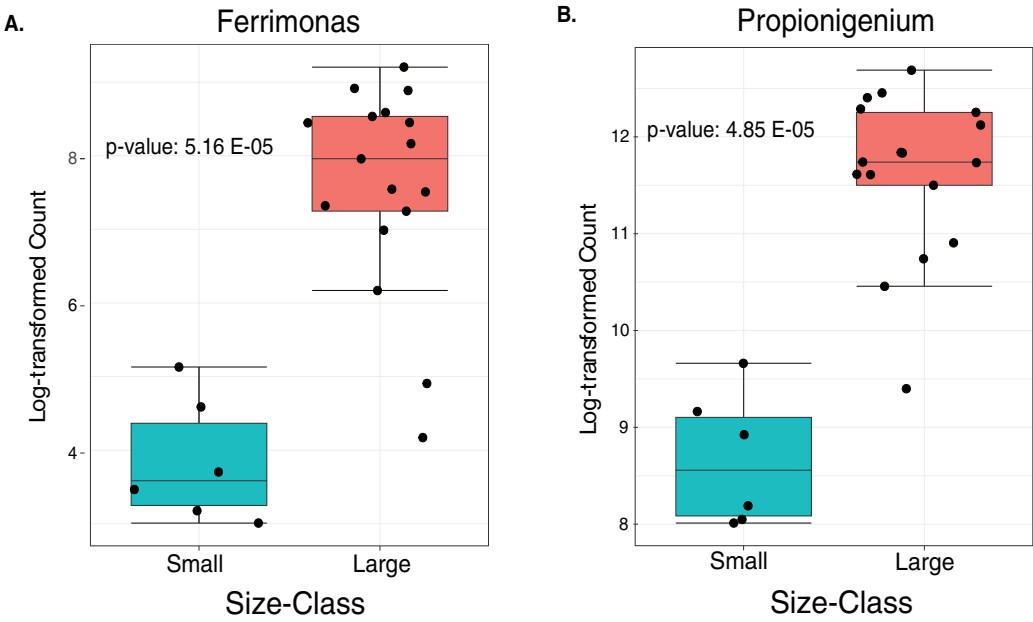

**Figure 5** **Boxplots of the two significantly increased genus-level plots for small and large sea urchins.** These plots correspond to LEfse analysis ($p \leq 0.05$) and include $n = 6$ small and $n = 35$ large animals (A and B, respectively).

The LEfSe analysis were visualized using box plots, showing that *Ferrimonas* ($p$-value = 5.15E−05) (Fig. 5A) and *Propionigenium* ($p$ = 4.85E−05) (Fig. 5B) were more dominant in the large sea urchins. A DEICODE resulting biplot revealed that the genera *Persicobacter* and *Spirochaetota* (g__V2072-189E03) had greater relative abundance in small urchins. In contrast, *Roseimarinus*, *Photobacterium*, *Desulfotea*, and *Propionigenium* had greater relative abundance in large urchins (Fig. 6A). The alpha diversity did not reach statistical significance, but small urchins exhibited apparently more diversity than large urchins ($p$ = 0.141) (Fig. 6B). Additionally, the observed features ($p$ = 0.52861), and faith_pd ($p$ = 0.44121) remained similar between size classes (Figs. 6C and 6D).

## DISCUSSION

This is the first study characterizing the gut microbiota of *T. ventricosus*, exploring the effect of water temperature and size class. This approach offers an in-depth understanding of the species' gut microbiota dynamics. The novelty of this manuscript lies in understanding how the fecal microbiota of the white sea urchin changes between size classes and in response to temperature changes in the sea urchin *Tripneustes ventricosus*. An initial characterization of the fecal pellet microbiota conducted during February, when water temperature is usually lower, was publicly available (*Rodríguez-Barreras, Tosado-Rodríguez & Godoy-Vitorino, 2021*); however, the fecal microbiota during August, when water temperature is higher, remained unknown. Therefore, we characterized for the first time the gut digesta microbiota for August and compared it with the February samples. Additionally, the fecal pellet microbiota of small individuals was characterized for the first time and compared with that of large sea urchins during the same time period, allowing us

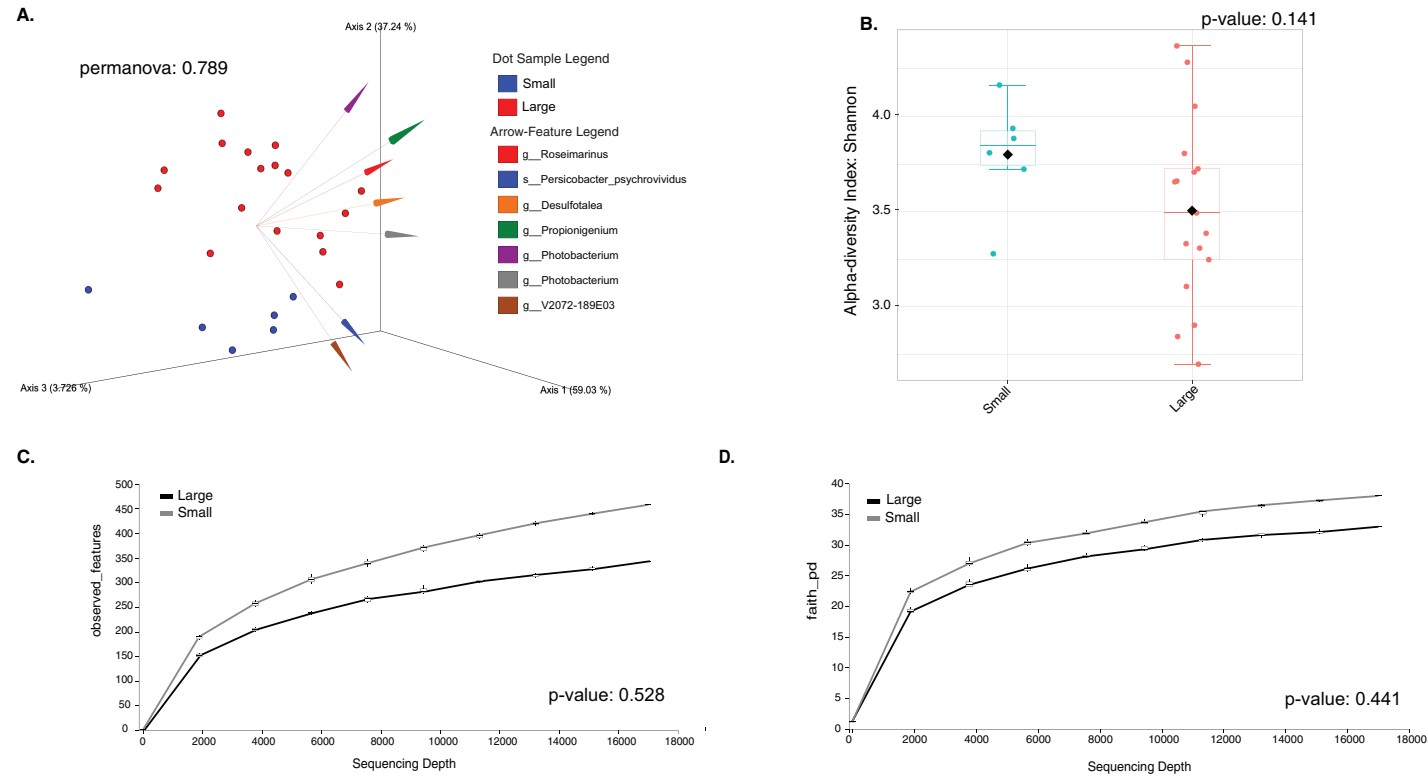

**Figure 6** **Beta and alpha diversity analyses of the gut microbiota in the sea urchin *Tripneustes ventricosus* according to size classification (small *n* = 6; large *n* = 35).** Beta diversity is represented in a 3D PCA biplot as a principal component analysis. Arrows corresponding to the specific feature (taxonomically characterized) and responsible for group clustering are colored. The arrows respond to Euclidian distance from the origin, and their size indicates the strength of the relationship of that ASV to the community composition and grouping (A). Alpha diversity estimatesare visualized by Shannon diversity boxplots (B) and rarefaction curves for observed features (C) and Faith's phylogenetic index (D).

to analyze how it changes between small and large sea urchins. The gut microbiota displayed changes in response to environmental fluctuation of abiotic parameters, like temperature more than by spatial variations, however, we consider this a big limitation in our study, as temperature and size evaluations are restricted to one sampling site and there are no variations in the animal sizes. Future studies should include a broader characterization of size classes and seasons to support any correlations. The composition and relative abundance of the gut microbiota in *T. ventricosus* during August and February were likely linked to annual changes in water temperature rather than pH or salinity. Water temperature changes were significant between February and August, while the other two measured abiotic parameters remained similar between both periods. The slight increase in alpha diversity found in this study during February could be related to bacteria genera being more evenly distributed in February than August. Nonetheless, we assume this is an important limitation of the study as other abiotic parameters such as pollutants (heavy metals, plastics, or chemicals that induce stress to the gut microbiota), water flow and currents (removing specific communities), differences in oxygen levels, or nutrient variations due to changes in diet (availability of macroalgae) (*Masasa et al., 2021*), which
may all account for changes in the gut microbiota of these invertebrates, and these factors remain to be evaluated.

Recent studies have reported differences in microbial communities across seasons, illustrating the effect of environmental factors (*Logue, Findlay & Comte, 2015*; *Karl et al., 2018*; *Ketchum et al., 2021*). Microorganisms from the Phyla Proteobacteria, Bacteroidetes, and Fusobacteria have been found to colonize the gut system in sea urchins (*Pagán-Jiménez et al., 2019*; *Faddetta et al., 2020*; *Feng et al., 2021*). In this study, we reported a higher relative abundance of the phyla Bacteroidota and Fusobacteriota and a reduction in relative abundance of the phylum Firmicutes, probably related to an increase in water temperature in August. Firmicutes have been previously found in high relative abundance associated with water algae and gut digest microbiota samples in the temperate sea urchin *Strongilocentrotus purtpuratus* at 13.1 °C (*Hakim et al., 2019*). This temperature is lower than those experienced by sea urchins in the Caribbean (Table S1), and probably water temperature could be the fact behind the reduction of Firmicutes reduction during February and indicates a less tolerance or lower performance of the groups at higher water temperatures. Indeed, a reduction in the relative abundance of Firmicutes was also detected with the increase in temperature in the gut microbiota of the bivalve *Mytilus coruscus* (*Li et al., 2018*). Another study reported a significant increase in the relative abundance of the Phylum Firmicutes from autumn to spring in the sea cucumber *Stichopus japonicus* inhabiting temperate waters (*Feng et al., 2021*). However, the lack of abundant studies in wild sea urchins makes it difficult to discuss the influence of water temperature. On the other hand, while the previously mentioned phyla experienced seasonal fluctuations in relative abundance, the phylum Proteobacteria remained relatively stable between February and August. However, this behavior was not observed in the intestinal microbiota of the sea cucumber *Holothuria scabra*, where Proteobacteria, especially the genus *Vibrio*, was found to have a higher relative abundance during the rainy season (*Plotieau et al., 2013*).

Seven genera experienced a reduction in relative abundance in samples collected in February. One of them was *Desulfotalea*, a sulfate-reducing bacteria known for forming symbiotic associations with invertebrates in anaerobic or sulfate-rich conditions (*Rabus et al., 2004*). They can contribute to the host's energy needs by metabolizing chemical compounds such as hydrogen sulfide. This genus is more abundant under lower water temperature, contrasting with our results (*Grim et al., 2023*). A potential explanation could be related with the fact that free-living species of *Desulfotalea* react in a different way to host *Desulfotalea* species, but also free-living species tend to be more affected by additional environmental factors such as the light incidence (*Grim et al., 2023*). A second genus that experienced a remarkable decrease from August to February in relative abundance was *Ferrimonas*. This genus is commonly found in aquatic environments and uses iron as an energy source through dissimilatory iron reduction (*Rosselló-Mora et al., 1995*; *Fan et al., 2013*). Warmer water temperatures led to an increase in the abundance of *Propionigenium*. Interestingly, another study found a similar pattern in *Tripneustes gratilla*, which was associated with the consumption of *Ulva* (*Masasa et al., 2021*).

On the contrary, at least five taxa increased in relative abundances in February. *Cutibacterium*, formerly known as *Propionibacterium*, is a Gram-positive bacterium commonly found in sebaceous areas of the skin (*Lee, Byun & Kim, 2019*). However, its presence in the samples may be due to human contamination and not because it constitutes a regular component of the microbiota of *T. ventricosus*. A second genus, *Haloferula*, was recently identified in samples from Sea Cucumber *Apostichopus japonicus* and its relative abundance associated with water temperature fluctuations (*Kang et al., 2023*). The candidate genus of small, cell wall-less bacteria in the Class Mollicutes, *Candidatus* Hepatoplasma, has been found in insects, specifically beetles, in their hepatopancreatic tissues (*Leclercq et al., 2014*). Candidatus Hepatoplasma are believed to be vertically transmitted and potentially mutualistic (*Leclercq et al., 2014*). While research has focused on the interactions with insects, a study in sea urchins has already established the detection of this taxa in sea urchins (*Hakim et al., 2016*). Limited information is available for the RF39 and MSBL3 strains. One potential seasonal factor associated with host microbiota shift could be nutrient availability. For example, a recent study demonstrated a high dynamisms level of the gut microbiota (*Bengtsson et al., 2024*), while other study found the ability of invertebrates to respond under seasonal changes in food supply spectrum (*Kivistik et al., 2023*).

Studying the gut digesta microbiota dynamic between small and large stages of wild caught sea urchins and other echinoderms offers valuable insights into the dynamic nature of these microbial communities during different life stages (*Clark & Walker, 2018*; *Carrier et al., 2021*). Previous studies have demonstrated significant changes in the gut microbiota during the transition from small to large urchins. However, few studies have been conducted addressing this issue in invertebrates (*Onitsuka et al., 2015*; *Miró et al., 2020*; *Popkes & Valenzano, 2020*), with limited research focusing on echinoderms (*Zhao et al., 2019*; *Carrier et al., 2021*; *Marangon et al., 2023*). Our findings revealed a size-related effect, providing insights into the progression of microbiota associated with different life stages. These studies concluded that small sea urchins exhibit higher microbiota diversity compared to adults, what agrees with our findings that also revealed that small *T. ventricosus* exhibited a slightly alpha diversity in their gut digesta microbiota compared to large urchins. This trend could be also related to significantly lower counts of *Ferrimonas* and *Propionigenium*. This difference in counts could be associated with the occurrence of more groups in small sea urchins. Particularly, *Propionigenium* tends to be a dominant group in adult sea urchins (*Yao et al., 2019*). Additionally, a higher relative abundance of *Ferrimonas* in adults could be related to the aestivation process and the reproductive cycle (*Kang et al., 2023*). A higher relative abundance of *Ferrimonas* has been found during this complex physiological process, which takes place during the summer season, coinciding with the period when our samples were taken. While large individuals invest more in reproduction, small individuals usually have undeveloped or absent gonads and therefore do not engage in reproductive activities (*Hendler et al., 1995*). Consequently, these two genera could serve as biomarkers for adult *T. ventricosus*.

The changes in the gut microbial community with size could be linked to feeding preferences. Literature has reported that sea urchins experience a dietary shift when

transitioning from small to large size classes (*Zann et al., 1987*; *Grosso et al., 2022*). Small *T. ventricosus* possess smaller mouths in contrast to large individuals, potentially explaining the differences in microbiota composition due to their ability to ingest different kind of particles. Furthermore, the aging process may lead to a gradual decline in biodiversity, favoring genera associated with immune responses and dysbiosis through evolutionary symbiosis (*Carrier et al., 2021*). A recent study conducted with the tropical sea urchin of the genus *Echinometra* sp., found differences among life stages, where small urchins exhibited a higher relative abundance of the Class Oxyphotobacteria (within the Phylum Cyanobacteria) compared to large urchins (*Marangon et al., 2023*). This finding agrees with our results, and other studies conducted in marine invertebrates where the microbial community display important changes across the animal life cycle (*Bernasconi et al., 2019*; *Quigley et al., 2020*). On the other hand, despite of lack of significant differences in alpha diversity found between small and large urchins, the slightly higher diversity found in small urchins is a pattern observed in sea urchins, related with natural transitions that occurs alongside life history (*Carrier et al., 2021*). Overall, taxa reported here such as *Psychromonas*, *Fusibacter*, *Propionigenium* or *Photobacterium*, can be considered keystones species in sea urchins as they were not only found in our study, but also in other sea urchin species (*Rodríguez-Barreras, Tosado-Rodríguez & Godoy-Vitorino, 2021*; *Ruiz-Barrionuevo et al., 2024*) as well as *Tripneustes gratilla* (*Masasa et al., 2021*) or *Lytechinus variegatus* (*Hakim et al., 2016*).

## CONCLUSIONS

Our study unravels the gut digest microbiota of *T. ventricosus*, focusing on the understudied aspects of seasonal and age-related dynamics, and underscores the importance of the gut microbiota of wild sea urchins and their potential associations with environmental variables. Comprehending the factors that influence gut microbial shifts is of utmost importance due to the significance of the microbiota in the overall function of the holobiont (*Pita et al., 2018*), being particularly critical due to rapid climate change (*Konopka, 2009*). Understanding the effect of temperature in gut bacteria will lead to valuable insights into these organisms' ecological and physiological adaptations to changing environmental conditions. Our findings suggest the existence of specific microbial profiles associated with different life stages in *T. ventricosus*, emphasizing the importance of life-stage-related factors in shaping the gut digesta microbiota. By demonstrating slight size-class changes in the gut digesta microbiota between small and large urchins, we highlight the dynamic nature of the host bacterial community throughout the animal's life cycle. By exploring the seasonal dynamics of the sea urchin gut microbiota influenced by fluctuating ocean conditions, and studying how microbial communities evolve from small to large urchins, we contribute unique insights guiding broader strategies for the conservation and sustainable management of coastal environments. Further studies should include a greater number of samples and collection sites to strengthen our capacity for drawing conclusions about *T. ventricosus* and to generalize to other similar sea urchin species in the Caribbean basin.

## ACKNOWLEDGEMENTS

We would like to thank the undergraduate students Jorge Hernández, María del Mar Fuentes, and Brayan Vilanova for their invaluable assistance in fieldwork and sample inventories. Hence, we extend our acknowledge to Mrs. Silvia Planas for her contribution to sequencing at the Sequencing and Genotyping Facility of the University of Puerto Rico (P20GM103475).

### Funding

This work was supported by the Puerto Rico IDeA Networks of Biomedical Research Excellence, Advancing Competitive Biomedical Research in Puerto Rico, 5P20GM103475-20 and the Center for Collaborative Research in Minority Health and Health Disparities (RCMI) U54MD007600. The funders had no role in study design, data collection and analysis, decision to publish, or preparation of the manuscript.

### Grant Disclosures

The following grant information was disclosed by the authors:
Puerto Rico IDeA Networks of Biomedical Research Excellence, Advancing Competitive Biomedical Research in Puerto Rico: 5P20GM103475-20.
Center for Collaborative Research in Minority Health and Health Disparities (RCMI): U54MD007600.

### Competing Interests

The authors declare that they have no competing interests.

### Author Contributions

- Ruber Rodríguez-Barreras conceived and designed the experiments, performed the experiments, analyzed the data, prepared figures and/or tables, authored or reviewed drafts of the article, and approved the final draft.
- Eduardo L. Tosado-Rodríguez performed the experiments, analyzed the data, prepared figures and/or tables, authored or reviewed drafts of the article, and approved the final draft.
- Anelisse Dominicci-Maura analyzed the data, prepared figures and/or tables, authored or reviewed drafts of the article, and approved the final draft.
- Filipa Godoy-Vitorino conceived and designed the experiments, performed the experiments, analyzed the data, prepared figures and/or tables, authored or reviewed drafts of the article, funding, and approved the final draft.

### Field Study Permissions

The following information was supplied relating to field study approvals (*i.e.*, approving body and any reference numbers):

Field collection was approved by the Department of Natural and Environmental Resources of Puerto Rico.

## Data Availability

The resulting 16S-rRNA sequences submitted to QIITA (González et al., 2018) under the Bioproject ID 12668;

The raw sequences are available at the European Nucleotide Archive: PRJEB40117 (ERP123720).

The QIITA project for the summer samples ID 13867 are available at the European Nucleotide Archive: PRJEB76415 (ERP160941).

https://www.ebi.ac.uk/ena/browser/view/PRJEB40117

https://www.ebi.ac.uk/ena/browser/view/PRJEB76415

## Supplemental Information

Supplemental information for this article can be found online at http://dx.doi.org/10.7717/peerj.18298#supplemental-information.

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
