# Peer review of "Effects of temperature and size class on the gut digesta microbiota of the sea urchin Tripneustes ventricosus"

_PeerJ, doi:10.7717/peerj.18298_

## Round 0.1 · original submission · Major Revisions

· Academic Editor

Major Revisions

The reviewers have provided a range of comments on the manuscript. Both reviewers generally appreciate the aims but have expressed significant concerns about the novelty of the study, along with the scope of discussion and citations of appropriate literature.

For further consideration, the manuscript will need to show substantial revisions. The authors should address all comments from the reviewers, paying particular attention to accuracy of statements made from cited references, and should make sure to cite appropriate literature in the field. This means re-writing large sections of Introduction and Discussion. The authors should be sure to address the comments about novelty and potential transience of the measured microbiomes. Please address carefully all comments about presentation and interpretation of results, and revise figures such that the text is legible from a printed page.

Additionally, please address these comments:
1. Replace web links with citations or company names, whichever applies.
2. Bray-Curtis dissimilarity should be referred to as such (not as ‘distance’) (L214).
3. Discussion should describe changes in relative abundances, not changes in abundances, since the latter were not measured (e.g. L304-305 and elsewhere if used).

**Language Note:** The review process has identified that the English language must be improved. PeerJ can provide language editing services - please contact us at [email protected] for pricing (be sure to provide your manuscript number and title). Alternatively, you should make your own arrangements to improve the language quality and provide details in your response letter. – PeerJ Staff

Reviewer 1 ·

Basic reporting

Rodríguez-Barreras and colleagues provide a series of comparisons using the sea urchin Tripneustes ventricosus to assess whether the gut microbiota is different between three sites along the coast of Puerto Rico, between juvenile and adult sea urchins, and between winter and summer. There is little that is particularly novel from this study, but the authors add to the growing evidence that sea urchins have a specific relationship with microbiota as well as that this microbiota can be influenced by the abiotic environment. The authors perform standard analyses to come to these conclusions, none of which raise any alarms. The only major issue that I found is that authors are not familiar with the echinoderm or microbiome/symbiosis literature, resulting in many inaccurate statements that are not supported by the literature that they reference. These details are below. I encourage the authors to carefully read their manuscript prior to submitting a revision, as it is currently has many cosmetic issues. Lastly, the authors are required to remake their figures as it is currently not possible to understand or read (!) them. An improvement here would allow for the reader to better appreciate this work.

Minor
Ln 36: change “genes” to “gene”, as the 16S rRNA is a single gene that has several variable regions. This correction should be made throughout the MS.
Ln 52: what is this lone “c” doing?
Ln 54: it is more technological than ecological
Ln 59-60: the connection to sea urchins is abrupt and not followed directly in the paragraph below. Please remove this or change to focus on gut microbiota.
Ln 68: if this is so widely observed, then it would be worth more than a single citation. Perhaps 3-5 from different taxa.
Ln 69: Reference is needed for ‘phenotypic plasticity’ claim. Also, how does this relate to temperature?
Ln 82: Dubilier does not concern echinoderms at all; this paper is chemosymbioses. I would reference multiple echinoderm-related studies that concern fitness or review the subject (e.g., Carrier & Reitzel, 2018, Nature Communications). This also applies to the rest of the paragraph and beyond.
Ln 84: Webster does not concern echinoderms at all; this paper is sponges. I would reference multiple echinoderm-related studies that concern immune and pathogens (e.g., Schuh et al 2020, Frontiers in Immunology; Ho et al 2016, Immunology & Cell Biology). This also applies to the rest of the paragraph and beyond.
Ln 86: completely wrong reference. Use: De Ridder C, Foret T. 2001 Non-parasitic symbioses between echinoderms. In Echinoderm studies (eds J Lawrence, M Jangoux). Abingdon, UK: Routledge. This is the only review that concerns this subject; no modern papers address it either.
Ln 90: Marangon concerned climate stressors. Schuh et al 2020, Frontiers in Immunology is appropriate here.
Ln 109-112: avoid ‘thus’ in multiple sentences in a row. Re-write.
Ln 103: provide details on the other comparisons (i.e., ‘2’ and ‘3’).
Ln 234-239: these are not results. Please remove.
Ln 240-241: provide all statistical tests and p-values here.
Ln 241-242: Alpha diversity is a two-dimensional measure of diversity. The first dimension concerns the number of unique ‘characters’ in the dataset, which can be assessed for microbiome-related amplicon datasets using total taxa and phylogenetic diversity. The second dimension concerns the frequency distribution of those characters, which can be assessed for microbiome-related amplicon datasets using measures of evenness and dominance. Each dimensional should be individually calculated and presented, and I request the authors do so. Please present OTUs (or, ideally, ASV) and phylogenetic diversity. Moreover, there are several estimates of alpha diversity (e.g., Shannon, Simpson, Choa1) that attempt to merge these two dimensions into one assessment and, in doing so, they do not fully represent the multi-dimensionality of alpha diversity; these should not be included. Please revise by assessing both dimensions in addition to the current analysis.
Ln 251: my alpha diversity comment applies here, as well.
Ln 267: "othe r" should be "other"
Ln 274: I am not sure one needs to compare the size between adults and juveniles, but the effort is noted.
Ln 279: the PERMANOVA should come before the differential abundance.
Ln 283-284: this does not make sense…

Figure 1 & 3: This text of all sections are not legible. A requirement prior to publication is that these figures need to be re-made for clarity and visual satisfaction.
Figure 2: This tends to be okay, but the overlapping text is a bit challenging. A legend is not needed, as the directionality is implied by the fold change.

Experimental design

see above.

Validity of the findings

see above.

Additional comments

see above.

·

Basic reporting

My concerns, comments and suggestions will be developed in the "General comments for the author" section.

Experimental design

My concerns, comments and suggestions will be developed in the "General comments for the author" section.

Validity of the findings

My concerns, comments and suggestions will be developed in the "General comments for the author" section.

Additional comments

This study examines the gut microbiota variability of the sea urchin species Tripneustes ventricosus in response to both seasons and to the development stage of the host. I recognize the intent of the authors to leverage a dataset associated with previously collected samples, reanalyzing it through a different perspective. This is a commendable approach, considering the ecological and ethical considerations associated with wild animal sampling. However, I have several major concerns and many specific comments about this work.
Main concerns:
- Mostly descriptive work, and so the discussion of the ecological implications for the host of the microbial diversity patterns described here is relatively poor.
- About my previous comment, I am not sure about the novelty of the work, as seasonal and age impact on the gut microbiota of sea urchins have been nicely demonstrated in the past. What is the originality of this sea urchin species?
- The absence of distinction (and also most likely the mixing) between gut content and gut tissue limits the conclusions. Indeed, how the author can be certain that they are not looking at the transient bacterial community, which probably does not have much impact on host ecology.
- Some statistical tests are missing to fully support the conclusions discussed in the manuscript (water temperature drives microbiota compositional shift in the sea urchin model, cf specific comments).
- The discussion should be better aligned with the study’s aims, avoiding tangential topics.
- The manuscript contains many typos and English can be significantly improved through a careful reading.
- The appropriateness of the references used should be reevaluated, as a substantial portion of them does not align with the corresponding statements.

Specific comments:

Introduction:
L52: « has been successfully c » cultured?
L53: the authors cited the work of Hugenholtz et al., 2009, yet it is unclear how this reference supports the statement that only 1% of the prokaryotic diversity has been cultured.
L59-60: The last part of the sentence sounds exaggerated, as available literature about sea urchin microbiota has significantly increased during the last couple of years, including temporal study (Ketchum et al. 2021). I’d suggest the authors to remove it.
The transition L72 and L73 needs to be smoothened (from temperature to host age). Remove the line break between the two paragraphs, as both deal with driving factors of host microbiota.
L74-77: Hard to follow, please rephrase.
L80-82: The reference cited does not mention echinoderms, nor host resilience. Please modify.
L82-84: Again, the reference cited is not about echinoderms. Please modify.
L84-86: Same comment as previously.
L89: first mention of the term ‘microbiome’, please define it, or stay with the term ‘microbiota’.
L90-93: There is no reference associated to echinoderms in this paragraph, so the statement is overly expansive. Moreover, the link between echinoderm microbiome and marine ecosystem equilibrium needs to be further detailed.
L94-L95: A bit contradictory with lines 59-60.
L105-109: Can be condensed.
L109-111: I would recommend the authors to make this statement less general. What are the predicted impacts of global warming on the ecological niche of T. ventricosus? What is currently known about microbiota response towards temperature fluctuation in other sea urchin models?
L114: “two categories” Which ones? There is only juvenile mentioned in the objectives.

Materials & Methods:
L131-133: How do the authors justify such a precise threshold? Please, specify in this section the statistical test used for comparing the two size categories.
L135-137: Unclear, please rephrase.
L142: “until it was attached to the surface”, please clarify and rephrase.
L142-143: and L140-141 can be merged.
L150: It means that gut tissue was also present in the processed samples. If so, you are also looking at the gut tissue bacterial community.
L147-151: can be resumed.
L154-161: please homogenize the wording (gut content or gut fecal pellets)
L56: remove “in this case”, as there is no other “case” presented.
L163: gut “content” samples.
L173: this sequence “dataset”.
L174-177: it is unclear whether the authors added new sequences to the previously published dataset.
L210-212: Unclear, please rephrase this sentence.
L214: What do you mean by ranked beta diversity? Plus, I would rather present the methodology related to alpha diversity first, and then the one associated with the beta.
L216-217: Did the PERMANOVA test includes a “strata” option (to perform the comparison between season within the same site)?
L220: “to establish connections” unclear.

Results:
L234-235: I have a major concern about the variability of the sequencing performance across the samples, as evidenced by the numbers reported in the supp table 3. For instance, the standard deviation provided for the condition Isla de Cabra/summer/5 adults is much higher than the average. Did the rarefaction at 17,000 reads lead to any sample removal?
L236-238: This should be moved to the mat&met section.
L238-239: Unnecessary, you could just state “adult” individuals in the following sentences.
L240-241: No difference in what?
L241: Cite Supplementary Figure 1a.
L244: Luquillo? Plus, cite Supplementary Figure 1c. And provided the statistical test result comparing Propionigenium abundances.
Supplementary Figure 1a is not described in the text. Moreover, be sure to rearrange the panel to fit the order of citation in the text. Are the relative abundances filtered?
L246-248: You did not test it. Besides, this sentence sounds like discussion.
L248: Uppercase is missing and the sentence is unclear.
L250: According to Figure 1A, Propionigenium is also discriminant of the summer sample.
L254: Please provide the statistical test result for each taxonomic group comparison and the associated values of relative abundances.
L257-259: Unclear and likely a copy-past. There is no clinical data here.
L257-261: Sounds like mat&met.
L261-262: I’m not sure to understand this sentence, MaAsLin2 was used to identify taxa at the genus level? Or to identify discriminating ASVs at the genus level?
L266: There is a typo mistake here.
L268-269: What do you mean? Move it to the Mat&Met section.
L272-274: the first part of this sentence should go to Mat&Met.
L274: Second part of the sentence; p-value in parenthesis.
In this specific result section, you should first demonstrate that there is no site effect on adult microbial community structure, before pooling it for comparison versus juveniles.
L281-282: Is this pattern maintained when comparing the juveniles with the adults of the same site (Isla de Cabra).
L283-287: Provide the p-values associated with the comparisons.

Discussion:
L294: This reference does not support the statement of the authors about marine ecosystem resilience, as it deals with the aquaculture of sea cucumbers.
L295: Again, the reference did not fit the statement. There is no demonstrated relation between compositional shift and host fitness in the cited work.
L298-299: You did not test it. And there are a lot of seawater properties not measured here that can drive this shift (e.g. primary production)
L299-300: Please also discuss the host seasonal physiologic variations or potential shifts in diet.
L307-L320: This paragraph is pretty disconnected from the results and the aims of the present work. This is a general discussion about the effect of seawater temperature increase on marine ecosystems in response to global change. Nothing to do with seasonality. Again, seasonality cannot be resumed at temperature conditions.
L322-325: Why this genus presence is decreasing in winter? How can the host benefit from bacterial-mediated sulfate-reduction? Moreover, you are looking (mostly) at gut content. So, the studied community might be likely transient in the intestines of the host, explaining the detection of generalist marine taxa such as Marinimicrobia. Overall, rather than discussing the ecological role of each genus, I would recommend the authors explain why these specific taxa might respond to the season.
L338-339: This can be omitted.
L339: A typo in the reference citations. The first paper from Elston et al. is missing in the bibliographic section.
L339-340: What do the authors mean here?
L336-354: same comment as for the previous paragraph. The authors should discuss the potential factors associated with the season that cause these changes rather than accumulating brief descriptions of each taxon.
L352-354: so, what were the main conclusions of these works? Are they consistent with the findings of the present study?
L356: Sela’s reference deals with the microbiota of freshwater insects. Doesn’t seem appropriate in the context of the sentence referring to echinoderms.
L361-362: what were the conclusions of these works?
L368: Through what mechanism?
L370-371: This should go in the introduction (if not already present).
L374: You did not cite this paper on line 362.
L374-376: You did not report a higher abundance of Oxyphotobacteria in your results, so how this result is consistent with your work?
L378-L379: the reference used is about sea urchins (not echinoderms in general). Please rephrase.

Conclusion:
L387: first use of the term “holobiont”. If not introduced in the introduction section, it needs to be removed.
L388: “gut content bacteria”. Moreover, direct evidence of temperature impact is missing.
L390: typos here. “Different life stages” in fact, there are only two in the study.
The term “life stages” should be used instead of “age” in the manuscript.
A finding cannot “suggest the detection”. Please rephrase.
L392: Another typo here “gutmicrobiota”.
L393: There was no significant difference in diversity.
L394-396: This statement is ambiguous and needs to be rephrased. The limited number of samples is a limitation for drawing conclusions about the specific species studied here.
In general, this conclusion section would need to be rewritten entirely to better fit the results and limitations of the present work.

Figure 1: Signification of the letter in arrow-feature legend is not provided. It is unclear whether arrows refer to discriminant ASVs or taxonomic groups. Please precise it in the legend. I would suggest the author reorder the figure panel to present first the taxonomic composition at the phylum level, then the alpha-diversity and finally the structure of the communities.
Figure 3: Homogenize the red and blue colour codes across the panel. Reorganize the panel as in figure 1.
Figures 3C and 3D: I would recommend the author filter the relative abundance data, to reduce the number of phyla represented in the figure, allowing the use of a more discernable set of colours. In addition, a supplementary table showing the means and standard errors of each phylum is required.
Supplementary Figure 1: Colors are barely discernable. Bray-Curtis is not an analysis, it is a distance method. Categories are not species as stated, but sites, please correct.
In 1C, please, change sample names with the codification provided in the manuscript.
In legend, “p-value” instead of “pvalues”. Also, precise the statistical test associated with the p-value.

---

## Round 0.2 · Major Revisions

· Academic Editor

Major Revisions

The manuscript has received another set of reviews from original reviewers. Both reviewers express frustration on the fact that their comments from the first round of reviews have not been thoroughly addressed. Neither of the original reviewers have completed a full review on this round because they felt the original issues had not been carefully addressed. Therefore, both the original reviewer comments and comments from the second round of reviews need to be addressed in a new revision.

A revision of the manuscript should be submitted with a new response letter which comprehensively address BOTH:

A) ALL first round reviewer comments + editorial comments and
B) ALL second round review comments + editorial comments.
Please provide a line-by-line response letter that includes A) followed by B).

Of particular note:

1. In the original review, Reviewer 1 provided a paragraph describing the major comments, but response to that part was not included in the response letter; it needs to be included and addressed. Both reviewers pointed out in the original review as well as the second round review comments that the narrative is not accurately or concisely reflecting existing literature. This point is critical to address, as is the revision of overall narrative development in the Discussion. See Reviewer 2’s comment about the importance of the discussion to focus on topics directly linked to the current study. This structural issue in discussion has seemingly not been addressed.

2. The authors have not addressed the comments from the editor nor PeerJ editorial office that were listed in the decision letter. One of the key issues is that the figures were and still remain unreadable from a printed page. Please split parts of the multi-panel figures to separate figures rather than trying to fit everything in a multi-panel, if the font becomes unreadable in the latter. Please refer to the original editorial letter decision for this and all the other comments from the editor and editorial office. These comments need to all be separately addressed in the response letter.

3. Many of the responses listed in the response letter are not accurate in describing what changes were incorporated in the revised version of the manuscript (see specific comments from both reviewers). Be sure to indicate accurately in the response letter how the point was addressed in the manuscript. It is strongly recommended that you state at what line(s) of the submitted track-changes Word file of the revised manuscript the incorporated change can be seen.

Please note that this invitation to revise does not guarantee acceptance.

Reviewer 1 ·

Basic reporting

see comments below.

Experimental design

see comments below.

Validity of the findings

see comments below.

Additional comments

I provided two major comments to the original submission by Rodríguez-Barreras and colleagues. The authors: (i) were clearly not familiar with the echinoderm or microbiome/symbiosis literature and (ii) were required to remake their figures because they were not possible to understand or read. Regarding the former, the authors merely pasted in the suggested references without removing the previous (and improper) references; this is sloppy and disheartening. Regarding the latter, the authors completely neglected this concern, so much so that they were directly disrespectful by deleting this text, while directly addressing all comments from Reviewer 2 prior to line-by-line responses. In my view: the authors addressed the small line-by-line comments by both reviewers and neglected the major comments; Figures 1 and 3 are illegible. For this reason, I do not feel that this manuscript has improved enough to fulfill the major revision that was required for publication and, thus, I cannot see the merit in publishing this manuscript in this current state.

·

Basic reporting

The manuscript version I received seems to be largely unchanged based on the comments provided below. Could you please verify if this is the intended version? Due to my uncertainty, I refrained from conducting an extensive review of the manuscript.

Initial comment: The absence of distinction (and also most likely the mixing) between gut content and gut tissue limits the conclusions. Indeed, how the author can be certain that they are not looking at the transient bacterial community, which probably does not have much impact on host ecology.
Author Response: We have added a statement discussing this fact in the discussion section.
This analysis portrays a description of the microbiota diversity associated with the animal’s gut, however we cannot define the differences between gut digesta, tissue-associated microbes and environmental transient bacteria. In fact, the host's capacity for positive selection might be a contributing factor in identifying and maintaining certain beneficial bacterial species while eliminating detrimental ones as discussed by Hakim et al, (2015). Only some bacteria from the environment are kept in the digestive tract (Harris,1993) as these ‘transient microbes’ can be acquired and excreted from the gut as feces.

I now understand that the author only analyzed the fecal pellets. To prevent any confusion among readers, it would be beneficial to provide a justification for this choice over the gut tissue in the introduction. Additionally, please ensure that the abstract clearly specifies that the analysis pertains to fecal pellets.

Initial comment: The discussion should be better aligned with the study’s aims, avoiding tangential topics.
Response: We have now improved our discussion throughout the text.

The author did not significantly improved the discussion (as illustrated by the presence of a very limited number of marked changes and various of my specific comments, see below).

Initial comment: L53: the authors cited the work of Hugenholtz et al., 2009, yet it is unclear how this reference supports the statement that only 1% of the prokaryotic diversity has been cultured.
Response: reference removed, and two new references added (Schleifer 2004; López-García & Moreira 2008).

The manuscript version I received is unchanged. Please correct.

Initial comment: L59-60: The last part of the sentence sounds exaggerated, as available literature about sea urchin microbiota has significantly increased during the last couple of years, including temporal study (Ketchum et al. 2021). I’d suggest the authors to remove it.
Response: Last part of the sentence was removed as suggested.

The manuscript version I received is unchanged. Please correct.


Initial comment: L80-82: The reference cited does not mention echinoderms, nor host resilience. Please modify.
Response: Reference replaced by Ho et al., 2016 and Schub et al., 2020.

Same comment as above.

Initial comment: L131-133: How do the authors justify such a precise threshold? Please, specify in this section the statistical test used for comparing the two size categories.
Response: The statistical test used for comparing size classes was specified. The result can be found in supplementary table S2. We also rewrote the sentences to clarify this issue.

Same comment as above.

Initial comment: L234-235: I have a major concern about the variability of the sequencing performance across the samples, as evidenced by the numbers reported in the supp table 3. For instance, the standard deviation provided for the condition Isla de Cabra/summer/5 adults is much higher than the average. Did the rarefaction at 17,000 reads lead to any sample removal?
Response: Yes, for the rarefication of 17,000 reads we removed 1 sample from adult and summer at one site (see Table 1)

I can’t find the table 1 in the files provided by the authors. Moreover, my concern about variability of sequencing performance has not been adressed.


Other specific comments :
L480 : replace « / » by « per ».
L481 : same comment as above.

L750-L751 : What the author mean by a « sizable dataset of high quality sequence reads »? Why such « high quality » ? Did the author apply any particular bioinformatic treatment to justify the use of this term ? Did they have any data to demonstrate this quality ?

L770-L773 : The authors are not discussing the result here. These lines sound like introduction section.

L775-L776 : « Recent studies » but there is only one work cited. Precise the host model.

The discussion of the seasonal changes in gut microbiota is week. Please, thoroughly compare with the work cited from Ketchum et al.

L782-L784 : Seasonal changes of temperature are not sudden. I do not understand the purpose of this sentence.

L783-L799 : The link of this paragraph with the study is unclear. The author characterized the seasonal effect, not an eventual eventual effect of anormal progresive increase seawater temperature nor heatwave effect. As a matter of fact, there is no precise result of the study discussed here, these lines are general considerations about the global warming effect on oceanic ecosystem. The author need to recenter this second paragraph of the discussion on their results.

L800 : Replace « During » by « In ».

L800-814 : Please rephrase the whole paragraph as it is hard to follow. The first line stated that seven genera were reduced (reduced in what ?), but Woesearchaeales and SAR406 are not genera.

L815 : Replace « feature » by « taxa ».

Experimental design

See above.

Validity of the findings

See above.

Additional comments

See above.

---

## Round 0.3 · Major Revisions

· Academic Editor

Major Revisions

1. The manuscript has been reviewed by the original two reviewers. One of the reviewers has provided additional feedback to improve the manuscript; specifically, providing two relatively major recommendations for improvement. I agree that separating temperature from season as a factor is an important point. Seasonal changes could be linked to a range of other factors besides temperature that were not measured.

2. The authors should also indicate clearly in the manuscript and the response letter what part of the manuscript is novel and adding to previously published work.

3. Previous comments from editorial office have not been separately responded to in the response letters. At least some of the issues still remain in the current manuscript. Please revise in the manuscript and explain in a new response letter how you have addressed all of the points listed below, including the new and previous editorial comments and the comments by the reviewer.


Version R2 (new editorial comments):

1. In the abstract and L290 you refer to Candidatus as a genus name. This is incorrect as Candidatus should be followed by a putative genus name. Based on Figure 1B and L378 you appear to be referring to Candidatus Hepatoplasma. Please correct.

2. Other line-by-line comments are below for a fair number of issues identified. Please address them and respond to each in your response letter.

L81 changing > changes?
L170-171 Delete ‘Once’
L201 You state that the data for your study species were published in a 2021 paper by the same authors. However, that paper only reports data for animals sampled in winter. Please clarify the information currently on L201. Were the summer data also published previously? If so, besides the sequence database information, please include citation to the paper.
L256 Bray-Curtis dissimilarity (not distance) (here and elsewhere in the text)
L288, 290, 291 greater relative abundance (not greater abundance)
L294-295 this sentence contradicts the sentence on lines 296-298. Please clarify.
L296-297 It’s difficult to concur with the reported trends in the Figure 2A. Why only Fusibacter in the winter, why not also Propionigenium, given their loadings were similar? Why is Bacteroidetes not mentioned as it seems to vary similarly to other taxa in the Axis 1 on the summer-winter difference?
L293 Supplementary Table 3 does not show results from PERMANOVA. Submitted Supplementary Table 3 is the same as Table 1. Please fix the numbering of Supp Tables 3 and 4 in submitted files and narrative. You are also referring to Figure 3 in Supp Table 4 legend – reference to Figure 3 seems incorrect.
L301 It appears you should refer to Figure 2D, not 2C with respect to Faith’s PD. Please correct and also explain the actual Figure 2C shown in the text.
L319 LEfSe
L322 Spirochaetota is not included in the Figure 6A. Is it the brown phylum with a letter code only as a name? Please clarify.
L324 If alpha diversity differences were not considered statistically different, one could not be reported as higher than the other. You could report it as an apparent but not statistically significant trend.
L334 You equate size to age throughout the manuscript. It would be appropriate to be explicitly accurate that you measured size, not age, and refer to size rather than age throughout (including the x-axis label in Figure 5, and any other tables and figures).
L373-374 I suggest you note that it is possible Propionibacterium could be a human contaminant
L400 delete ‘in’
L402 and 403 ‘slightly diversity’ stated twice. It appears a word is missing twice.
L441 clarify: ‘slightly life-stage changes’
Table1. Omit decimals from read numbers.
Figure 6A. Permanova is reported as 0.789. The Permanova result in these PCOA analyses is confusing, because the methods description didn’t associate this test with this analysis. Permanova analysis for size shows as 0.005 in the Supplementary Table, instead of 0.789 reported in Figure 6 and L314-315. Why are the numbers different?

--
Previous editorial comments. First review (R0) editorial comments requiring a response:

For further consideration, the manuscript will need to show substantial revisions. The authors should address all comments from the reviewers, paying particular attention to accuracy of statements made from cited references, and should make sure to cite appropriate literature in the field. This means re-writing large sections of Introduction and Discussion. The authors should be sure to address the comments about novelty and potential transience of the measured microbiomes. Please address carefully all comments about presentation and interpretation of results, and revise figures such that the text is legible from a printed page.

Additionally, please address these comments:
1. Replace web links with citations or company names, whichever applies.
2. Bray-Curtis dissimilarity should be referred to as such (not as ‘distance’) (L214).
3. Discussion should describe changes in relative abundances, not changes in abundances, since the latter were not measured (e.g. L304-305 and elsewhere if used).

**Language Note:** The review process has identified that the English language must be improved. PeerJ can provide language editing services - please contact us at [email protected] for pricing (be sure to provide your manuscript number and title). Alternatively, you should make your own arrangements to improve the language quality and provide details in your response letter. – PeerJ Staff

Reviewer 1 ·

Basic reporting

good

Experimental design

good

Validity of the findings

good

Additional comments

good

·

Basic reporting

I thank the author for clarifying my previous concerns about the manuscript and for their work on this new version. Unfortunately, I still have two (relatively) major comments regarding this new version:

Temperature and size class analysis:
The authors detected a significant difference in temperature between the two seasons. Independently, they performed a Lefse analysis using the "season" factor and suggested that the significant variations detected in bacterial taxa abundance are linked to the previously evidenced differences in seawater temperature.
While it's somewhat obvious that seawater temperature is higher in summer than in winter, I believe a correlation test between temperature values and specific taxa abundances would be more appropriate. Although the number of values in the case of the temperature may limit the analysis, the varying and specific sizes of the specimens analysed could allow the detection of robust correlations with specific bacterial taxa over host size.
Moreover, it's important that the authors acknowledge an important caveat of the manuscript: only three environmental conditions were measured at a very specific point in time. Additional evidence could be provided to further support the idea that temperature is the most discriminant factor between seasons in this specific geographic area (using an environmental database such as BioOracle). In any case, I would suggest that the authors propose alternative scenarios to explain the differences, and the discussion needs to be restructured to avoid mixing season and temperature.

Novelty and Data Concerns:
My second major concern remains (as mentioned in my previous revision) the apparent limited novelty of the work. All the sequencing data have already been published in the same journal with no new analysis except for the associated metadata. The experimental design and implemented analysis do not allow for a full understanding of the underlying factors of the detected patterns (only three environmental variables measured, as noted in a previous comment, and the variability of host sizes is reduced to only two classes). As a result, the work is mostly data-driven (recycling of past data) without a working hypothesis. In its current form, the manuscript presents a low diversity in the analysis performed, relying exclusively on composition, which greatly limits the discussion of the observed patterns. Indeed, the current version of the discussion is highly dichotomous (absence or presence of difference in a few ASV abundances). I recommend that the authors further complement the work by integrating, for instance, co-occurrence analysis, which could help unravel the significance of the few discriminant ASVs across size classes and seasons in the microbiome structure of T. ventricosus.

Introduction:
- Line 70: Please remove "and pollution" as it is included in environmental condition changes.
- Lines 75-76: Specify the kinds of changes and provide references, ideally in sea urchin models, or remove the redundant sentence.
- Lines 91-109: This paragraph needs to better connect to the work's objective (seasonal and age effects on T. ventricosus microbiota).
- Line 105: Remove the extra period before the citations.
- Lines 111-123: Clarify why this is a good model to study seasonal and age-related variations in gut microbiota.
- Lines 124-132 and whole introduction: The structure still suggests a data-driven rather than hypothesis-driven approach. Clarify the novel hypothesis of the research and why this sea urchin species is a good model.
- Lines 124-126: The latter part of this sentence is redundant.
- Lines 127-128: This sentence is too general; please remove or relocate it.
- Line 132: Introduce the "size class" concept earlier when presenting the model's advantages.

Methods and Results:
- Line 263: Distinguish between "age" and "life-stage" as there is no relation between the two factors in the manuscript.
- Lines 299-301: Standardize and reduce the number of decimals in the p-values.
- Lines 301-308: Present the numbers of discriminant OTUs in relative abundances, e.g., "Specifically, only 4% of the taxa (i.e., 7 ASVs) exhibited a significant decrease [...]". Provide the relative abundance of these taxa in the whole gut community to better illustrate the seasonal change importance within the microbiota of T. ventricosus. The changes appear globally limited. Have these taxa been previously identified as potential keystone species in the gut microbial community?
- Line 325: Specify which "other features" are being referred to.
- Line 354: Explain the stability compared to sea cucumber.
- Lines 341-361: Consolidate mentions of Firmicutes into a single section of the discussion.
- Lines 351-352: In which environmental context this reduction has been observed in the host?

Figures and Tables:
- Figure 1: In the last sentence of the figure caption, ‘summer’ appears twice.
- Lines 545-546: The spacing needs to be homogenized.
- Lines 626-627: Standardize the font of the citations.
- Figure 2: Remove uninformative taxonomic information in the Arrow Feature Legend.
- Figure 4: Add a space in the title. Order the legend alphabetically and place the "other" category at the end of the legend key.
- Figure 6: Correct typos in the title and legend (spaces are missing). In panels C and D, start the x-axis at 0 instead of -2,000 sequences.
- Figure 5: The boxplots present only the LefSe results. Generate a similar figure for markers identified through DEICODE.
- Supplementary Table 1: The legend is truncated ("[...] significant between?") and needs rephrasing.

Experimental design

Cf basic reporting section

Validity of the findings

Cf basic reporting section

Additional comments

I am disposed to provide further input to strengthen this work.

---

## Round 0.4 · accepted · Accept

· Academic Editor

Accept

Please correct the following in the final version while in production:

***The line numbers refer to line numbers in the submitted track changes version:***

L24 varied > varies
L32, 34 more abundant > relatively more abundant
L39 abundance > relative abundance
L85 other > another
L367 Given you refer to alphadiversity, it seems you should be referring to Supplementary Figure 1C here. Do you actually mean betadiversity?
What test is the p=0.696 referring to? Please add the name of the test.
L375 ‘in collection sites’: do you actually mean: ‘among collection sites’?
L383 v > vs.
L384 vs > vs.
L385 abundance > relative abundance
L387 in Candidatus Hepatoplasma, remove italics from ‘Hepatoplasma’
L388 were more abundant in February > had greater relative abundance in February
L390 structure > composition
L390 p-value > p (here and anywhere else where used)
L419 increase > increase in relative abundances
L432, 433 was/were more abundant > had greater relative abundance
L465 onwards; the first sentence of discussion: Please modify grammar and message of the sentence based on your results >> ‘…characterizing the gut microbiota of T. ventriculosus, exploring the effect of season and size class’.
L566 affected for > affected by
L571-572 sentence is unclear